# ProtFIM: Fill-in-Middle Protein Sequence Design via Protein Language Models

## Abstract

Following the investigation that protein sequence determines its structure and function, engineering protein sequences allows us to optimize the functions of proteins for specific purposes such as enhancement of catalytic activity or binding affinity maturation. In protein engineering, there are many cases where the amino acids in the middle of a protein sequence are changed while maintaining the remaining residues to avoid unwanted functional changes from remaining residues. However, existing research on protein sequence design via protein language models (pLMs) has focused on modifying suffix residues by prompting prefix residues to the model or mutating the overall sequence residues. This is unsuitable for scenarios where the residues located in the middle of the sequence are to be optimized. In this work, we suggest a pLM-based framework to solve the fill-in-middle (FIM) protein engineering tasks. To evaluate the performance of pLMs on the FIM tasks, we design a novel evaluation scheme where pLMs are tasked to generate new sequences while maintaining the secondary structures. Also, we propose a new PROTein language model specialized for the Fill-In-Middle task, Prot-FIM. Experiments confirm that ProtFIM performs FIM engineering efficiently, especially for alpha-helix structures, and provides decent protein representations of sequence-function relationships. Finally, we demonstrate an artificial protein sequence design framework composed of ProtFIM and a high-quality structure predictor as a novel tool to optimize protein sequences.

## 1 Introduction

Proteins play a crucial role in various parts of biological processes, and the ensemble of diverse functioning proteins is the basis of life's activities, such as immune response and metabolism. Such essential and versatile functions of proteins are encoded in protein sequences which are the arrangement of amino acid residues. The sequences determine their structures via complex biophysical interactions between residues and these structures are directly linked to the functions of proteins. Thus, optimizing the protein's function by changing amino acid residues of protein of interest, called protein engineering, has been of great interest in diverse industries such as biofuel (Wen et al., 2009), pharmaceuticals (H Tobin et al., 2014), and agriculture (Rao, 2008).

One of the representatives of protein sequence design methods is a mutagenesis technique, which gives evolutionarily plausible candidate protein sequence libraries with the help of genetic engineering (Arnold, 1998). However, this approach requires substantial efforts in high-throughput screening experiments. Recently, machine learning-guided protein sequence design strategies have been proposed to achieve a more efficient sequence space search using experimentally acquired labeled data (Yang et al., 2019a).

With both advances in high-throughput sequencing technologies and language modeling in the field of natural language processing (NLP), protein language models (pLMs), which are trained in an unsupervised manner using tremendous sets of unlabeled protein sequences (Consortium, 2019), have been developed for generating *de novo* protein sequence (Madani et al., 2020; Hesslow et al., 2022; Moffat et al., 2022; Ferruz et al., 2022; Nijkamp et al., 2022). Existing generative pLMs are trained using an auto-regressive (AR) strategy (Radford et al., 2019; Brown et al., 2020), and generate sequences conditioning on the prefix protein sequences. Unfortunately, if the target region where we want to change amino acid residues is located at the front, existing pLMs uses only

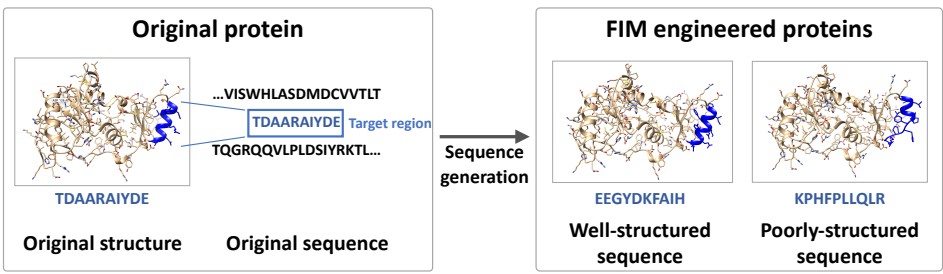

Figure 1: An illustrative example of FIM protein engineering. The changed sequences for the target region are generated by generative pLMs or the like, and the structures are altered accordingly.

a few preceding amino acid residues ("prompts") for sequence generation. The interaction sites, positions that interact with other proteins or molecules to perform their functions and are mainly modified to improve functionality, are evenly located on the protein sequence. To prove this, we collect 3D protein structures from Protein Data Bank (PDB) database (Sussman et al., 1998) and calculate the relative positions of protein-protein interaction sites on the protein sequences (see details in Appendix A.1). As illustrated in Figure 5, interacting sites are evenly present on the protein sequence. This result suggests that in protein engineering, modifying the amino acid sequence will be done for the middle part of the sequence in many cases. In this case, existing pLMs may not effectively utilize the information behind them, which can result in poor quality of generation.

In this work, we regard the middle protein engineering as a fill-in-middle (FIM) sequence generation problem as in Figure 1 and investigate the possibility of pLMs in FIM protein engineering framework. With the emergence of highly accurate protein structure predictors (Jumper et al., 2021; Baek et al., 2021), protein structures are predicted very quickly and accurately with a low cost. Using these advances, we propose a new evaluation scheme, Secondary structurE InFilling rEcoveRy, SEIFER, for FIM protein sequence generation. The secondary structures are usually desirable to be preserved (Rubio et al., 2019) since the binding pockets of other interacting proteins or molecules are fixed to some extent. In SEIFER, models are tasked to recommend protein sequences and achieve two conditions: the new sequences must be different from the original sequences and their secondary structures must be fully maintained. So, SEIFER can assess both the diversity and structure of new sequences simultaneously and we believe that SEIFER is suitable for assessing generated sequences in the field of protein engineering which modifies the amino acid residues of original sequences to improve functions. Also, inspired by the latest research in the field of language models (Bavarian et al., 2022b), we propose a new Protein language model specialized for the Fill-In-Middle task, ProtFIM. Compared to existing pLMs, our proposed ProtFIM use both front ("prefix") and back ("suffix") sequence information during training and inference.

Through SEIFER evaluation, we show that ProtFIM can generate diverse sequences while maintaining secondary structure, especially for $\alpha$-helix. Furthermore, ProtFIM outperforms when engineering on residues positioned in the front part of a protein sequence compared to existing pLMs, proving that the FIM training is more suitable for FIM engineering compared to AR pLMs. Finally, through analysis and visualization, we prove that ProtFIM has decent representations of protein sequences and can serve as a sequence optimization tool accompanied by AlphaFold2. In summary, our contributions are:

- We define FIM protein engineering as protein sequence infilling tasks and provide the applicability of protein language models on the task.

- We propose a new evaluation scheme, SEIFER, that can be used to evaluate the performance of pLMs on protein infilling sequence design tasks by considering structural conservation. Through this evaluation, we find that existing AR pLMs are capable of sequence design having $\alpha$-helix structure.

- We propose a new type of pLM, ProtFIM, that has both AR and FIM capability. Comprehensive results show that ProtFIM has efficient and comparable performances in protein infilling and protein representations for protein engineering compared to other pLMs.

- We show that the ProtFIM acts as a sequence optimizer, which generates novel sequences with high pLDDT of AlphaFold2 while maintaining the structures essential for the function of the protein.

## 2 RELATED WORK

**Protein language models** Pretraining-based language modeling such as Transformer (Vaswani et al., 2017), BERT (Devlin et al., 2018), and GPT (Ferruz et al., 2022) have revolutionized natural language processing and shown remarkable performance on various tasks such as language under-standing, sentence generation, and infilling over the last few years. With a huge increase in the amount of unlabeled protein sequences (Consortium, 2019) produced by high throughput sequencing technologies, pLMs have been introduced and resolved the challenges in protein science and engineering by learning protein languages. BERT-style models primarily provide protein embeddings to solve prediction problems, including protein structure prediction (Rao et al., 2020; Jumper et al., 2021; Lin et al., 2022), function prediction (Brandes et al., 2022), and property prediction (Rives et al., 2021). GPT-style architectures are utilized in resolving generation challenges such as protein sequence design (Madani et al., 2020; Hesslow et al., 2022; Moffat et al., 2022; Ferruz et al., 2022; Nijkamp et al., 2022).

**Protein sequence design** Attempts to efficiently design protein sequences can be divided into two categories: a method for conducting a large number of high-throughput experiments with mutagenesis and a machine learning-based sequence generation method. Recent advances in experimental-based methods (Fowler & Fields, 2014) allow us to assess the functional changes of mutated protein sequences at a large scale and produce a lot of labeled data. Many machine learning-based sequence design methods generate the optimized sequences iteratively based on the feedback of labeled data (Yang et al., 2019a; Xu et al., 2020; Wu et al., 2021; Shin et al., 2021). Unfortunately, both approaches require a lot of cost and effort in experiments. Recently, several works generate protein sequences conditioned on given 3D structures using a single energy function (Alford et al., 2017), convolutional neural networks (Zhang et al., 2020; Qi & Zhang, 2020), graph neural networks (Ingraham et al., 2019; Jing et al., 2020; Strokach et al., 2020; Dauparas et al., 2022), or Transformers(Hsu et al., 2022). Since these works require 3D coordinate information to generate sequences, generation may be limited only to areas where high-quality structures exist. Also, in these works, CATH (Orengo et al., 1997) is used to evaluate how similar the generated sequences are to the original sequence. This evaluation method may not be suitable for protein engineering, which aims to change the sequence to have a better function. In parallel, generative pLMs such as RITA (Hesslow et al., 2022), DARK (Moffat et al., 2022), ProtGPT2 (Ferruz et al., 2022), and Progen2 (Nijkamp et al., 2022) have been developed. These generative pLMs generate protein sequences having well-folded and viable structures even though these methods do not employ any structural information. However, due to the nature of the AR model itself, these methods utilize only the preceding sequence information during sequence generation. Our proposed ProtFIM has both AR and FIM property, resulting in efficient FIM protein engineering.

## 3 METHOD

**Problem Setup** In NLP, infilling is defined as generating complete text $x$ given incomplete text $\tilde{x}$, including one or more missing spans. Similarly, we can regard protein engineering on middle residues as an infilling task where models are tasked to return new protein sequences $s$ given incomplete protein sequence $\tilde{s}$ containing missing residues on the target region. Additionally, in the protein infilling task, there is a special structure conservation constraint where the secondary structure of the target site is maintained to approximate the protein engineering scenario properly. Taken together, our goal is to develop a pLM, $f(\tilde{s}; \boldsymbol{\theta})$, which outputs complete protein sequence $s$ based on a distribution $p(s|\tilde{s})$ and sequence $s$ must have different residues while having the same secondary structure as that of original residues.

### 3.1 MODEL REQUIREMENTS

We suggest four key characteristics of pLMs suitable for protein infilling tasks as follows:

- Dynamic property: The model can handle various lengths of protein sequences because the lengths of the middle sites are diverse depending on various applications.

- Causal modeling: Previous studies reveal that AR pLMs have data-driven co-evolutionary rules across natural protein sequences and generate plausible sequences that tend to be well-folded. So, pLMs which have both AR and infilling capability would be optimal.

- Efficiency: Various strategies, such as pre-processing training data, modifying the model architecture, and using special tokens for controlling, can be used. However, these approaches must be fulfilled as efficiently as possible because protein sequence length is relatively long (we use the maximum length of residues as 1024 in this work).

- Diversity: Because there are many combinations giving the same secondary structures, pLMs which generate diverse sequences different from existing sequences are preferred.

To achieve the above characteristics, we adopt the idea of FIM transformation, which is a very recently proposed FIM causal language modeling strategy by Bavarian et al. (2022a). The following section explains how to develop FIM pLMs and generate protein sequences using the model.

## 3.2 MODEL DEVELOPMENT

**FIM training** In FIM transformation, a span of text from the middle of a whole sentence is moved to its end, and additional special tokens are introduced for marking where spans are from. The transformation is stochastically fulfilled during causal language modeling training. Intriguingly, this simple and straightforward transformation successfully gives fill-in-the-middle (FIM) capability to the model without modifying model architecture and sacrificing left-to-right causal generation capacity. The transformation is easily applied to protein sequence modeling as follows. First, we tokenize each residue $R$ of a protein sequence $S$ with length $N$ to the sequence consisting of corresponding tokens $T$ (see eqn. 1 and 2).

$$S = (R_1, R_2, ..., R_N) \tag{1}$$

$$S_t = (T_1, T_2, ..., T_N) \tag{2}$$

Second, we conduct uniform sampling to get the start position $K$ of the middle span of length $L$ and add special tokens [PRE], [MID], and [SUF] at the beginning of each prefix, middle, and suffix part, respectively. Finally, FIM-transformed sentences are created by concatenating prefix, suffix middle in order as eqn. 3.

$$S'_t = ([PRE], R_1, ..., R_{K-1}, [SUF], R_{K+L+1}, ..., R_N, [MID], R_K, ..., R_{K+L}) \tag{3}$$

Because several residues are needed to form a secondary structure, the middle residue sampling is conducted so that both prefix and suffix parts have at least four residues. The traditional GPT2 architecture from Hugging Face (Wolf et al., 2019) is used for training, and FIM transformation is applied to the input with a 50% frequency. We denote pLMs trained using FIM transformation as ProtFIM in this work. More details are written in Appendix A.4.

**FIM inference for middle residue engineering** For generating complete sequences in protein infilling tasks, we consider the target region as the middle part, and the front and back regions to the target region are prefixes and suffixes. Then, we make a prompt for FIM generation by concatenating prefix part, suffix part, and [MID] token as eqn. 4.

$$P'_t = ([PRE], R_1, ..., R_{K-1}, [SUF], R_{K+L+1}, ..., R_N, [MID]) \tag{4}$$

## 4 EXPERIMENTS

Section 4.1 illustrates our proposed evaluation scheme, SEIFER, specially designed for protein infilling tasks. Section 4.2 describes metrics and various baseline models covering representative language modeling approaches such as causal language modeling (CLM) and permutation language modeling (PLM). Section 4.3 includes evaluation results of SEIFER tasks. Then, section 4.4 and 4.5

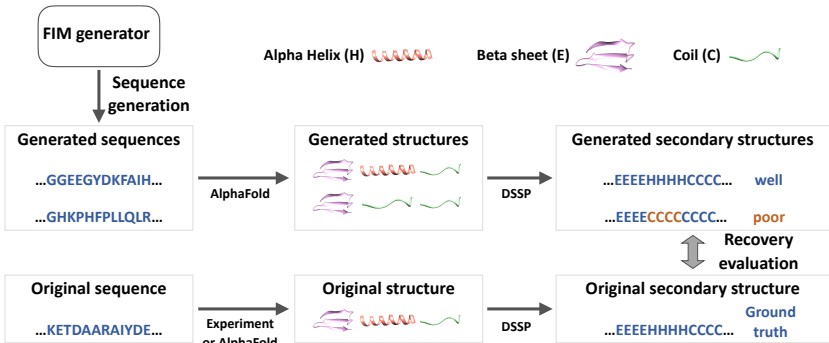

Figure 2: Illustration of our SEIFER evaluation scheme, which estimates the recovery rates of the secondary structures of the generated structure for the original secondary structure.

provide ablation studies of the SEIFER task concerning relative position and length of target region. Additionally, other metrics for evaluating pLMs, such as perplexity and sequence recovery, are provided in AppendixA.6.

## 4.1 EVALUATION

**Protein secondary structure**   Protein secondary structures play a key role as an intermediate between the primary sequences and functional tertiary structures that determines the function of proteins in a variety of biological process. Therefore, designing properly optimized combinations of residues having the same secondary structures can lead to enhanced function of protein (Rubio et al., 2019). Protein secondary structures are categorized into regular and irregular categories. First, the regular structure includes $\alpha$-helix (H), $\beta$-sheet (E) (Pauling et al., 1951), and the irregular structure type is a coil (C). In this work, we adopt a 3-class secondary structure definition and those structures are calculated via DSSP (Kabsch & Sander, 1983).

**Protein secondary structure recovery via infilling**   In this work, we propose a new evaluation scheme, Secondary structurE InFilling rEcoveRy, called SEIFER, evaluating the sequence generation and structure conservation simultaneously. In the task, first, models are tasked to generate various sequences to fill the provided target sites. Since secondary structures are calculated based on three-dimensional structural information, the characterization of tertiary protein structures for each generated sequence must be preceded. Unfortunately, conducting experimental characterization on all the new sequences is practically impossible. Instead of this, we utilize Alphafold2 (Jumper et al., 2021), which has shown near-experiments prediction performance, to predict tertiary structures of all generated sequences. Then, secondary structures of each new sequence are calculated via DSSP algorithm using DSSP module of Biopython (Cock et al., 2009). Finally, the secondary structures of new sequences are compared to the original secondary structures. We assign a positive value, 1, on the case where all new residues have the same secondary structure as the original sequences. And all other cases are negative, 0. We illustrate the process of SEIFER in Figure 2. We use proteins presented in CASP14 to obtain candidate middle sites for SEIFER tasks. And, we argue that our experimental setting is reliable because AlphaFold2 was stringently assessed and proved by remarkable prediction performance on the proteins in the CASP14. Additionally, we use the middle sites, which have minimum lengths of 10, 6, and 6 for helix, beta, and coil structures, respectively, considering the average number of residues for the structures.

**Difference of SEIFER over protein residue recovery task**   Sequence recovery has been widely used to evaluate the generation performance of protein generative language models (Ingraham et al., 2019). However, considering that the objective of sequence optimization is to design new sequences with better target properties, recovery of original residues would not be proper. So, a metric is needed to evaluate whether the model can generate a variety of sequences while maintaining the function of the protein. Because the function of protein is directly linked to local structure, evaluating the model ability that generates different residues with the same local structure is a promising way. So, we argue that our proposed SEIFER tasks are appropriate for simulating sequence engineer-

ing scenarios, because in SEIFER tasks models are tasked to recover the protein's local structure, especially, secondary structures, not residues.

## 4.2 EXPERIMENTAL SETUP

**Baseline**    We compare our ProtFIM with other pLMs covering diverse generation strategies.

- ProGPT2-C: To prove the effect of FIM over AR in protein engineering, we train AR-based pLMs using the same data, hyperparameters, and the number of parameters compared to ProtFIM. It is similar to the previous AR model, ProtGPT2 (Ferruz et al., 2022), but our model is trained in the amino acid level. Thus, we name the amino acid residue-level (character-level) version of ProtGPT as ProtGPT2-C.
- ProGen2: ProGen2 (Nijkamp et al., 2022) is a concurrently released suite of AR pLMs with various parameters.
- ProtXLNet: XLNet(Yang et al., 2019b) can protein sequences using prefix and suffix information. Like FIM, sequences of target sites are generated auto-regressively with conditioning on bidirectional context from both the prefix and suffix. We borrow the publicly released ProtXLNet model, a variant of XLNet for protein (Elnaggar et al., 2022).
- Random generator: This generator is used to approximate the random mutagenesis technique, error-prone PCR (McCullum et al., 2010), which is still commonly used in protein engineering (Dror et al., 2014).

**Evaluation metrics**    SEIFER measures how many sequences with the same secondary structure exist among new sequences created by a generative model. It is like a retrieval or recommendation engine for protein sequences. In the SEIFER task, all models generate K sequences for N middle sites, and all sequences are evaluated by whether the whole secondary structures at each target site are recovered. If the whole secondary structures are recovered, it is a true positive (TP). Then, Precision@K is the mean of TP/K for N sites. Also, we use Retrieval@K, which assumes a positive case where any true positive sequence exists in generated K sequences, zero otherwise. Thus, Retrieval@K is (the number of sites having TP among K)/N.

## 4.3 EXPERIMENTAL RESULTS ON SEIFER TASKS

As shown in Table 1 and  2, all pLMs have better performance than the random generator in helix structure recovery in views of both retrieval and precision. These results describe that pLMs are promising tools to fill in middle residues of target protein during protein engineering on helix structure. In contrast, all pLMs perform similarly or worse than the random generator in the $\beta$-sheet and coil recovery. To investigate the result, we check the distribution of secondary structures for the proteins with known structures by calculating the distribution of secondary structures in proteins from PDB (details are described in AppendixA.2). Figure 6a and 6b illustrate 3-classes and 4-classes secondary structure distribution, showing that the $\alpha$-helix structures are dominant in natural protein structures. This empirical result is consistent with the widely known observation in the protein community. We conjecture that this imbalance gives unwanted $\alpha$-helix bias in existing protein sequence datasets. Additionally, the coil usually has unordered noisy structures. Taking the above facts together, it is possible to say that the similar or worse performances of pLMs in $\beta$-sheets and coil cases are reasonable because helix bias makes it models hard to learn the rules of generating residues consisting of the coil and $\beta$-sheet.

Meanwhile, we compare ProtFIM over ProtGPT2-C to see the effectiveness of FIM compared to CLM. As shown in each table's fifth and sixth rows, ProtFIM performs better than ProtGPT2-C in helix recovery. Because both models are trained using the same data, hyperparameters, and the number of parameters except for the utilization of fill-in-middle transformation, these results support that conditioning on both prefixes and suffixes during generation is essential for better sequence design for protein engineering.

We also compare ProtFIM with other pLMs, such as ProGen2 and ProtXLNet. XLNet is another possible model which is able to fill-in-middle protein engineering using both the prefix and suffix. It is found that ProtXLNet shows strong performance in $\alpha$-helix compared to the similar scale of

Table 1: Model performances on SEIFER tasks in terms of retrieval.

| Model | #Params | Objective | H ($\alpha$-helix) | | E ($\beta$-sheet) | | C (Coil) | |
|---|---|---|---|---|---|---|---|---|
| | | | R@3 | R@5 | R@3 | R@5 | R@3 | R@5 |
| Random Generator | - | - | 0.46 | 0.60 | 0.76 | 0.82 | 0.76 | 0.80 |
| ProGen2-small | 151M | CLM | 0.59 | 0.71 | 0.67 | 0.64 | 0.69 | 0.80 |
| ProGen2-medium | 764M | CLM | 0.54 | 0.66 | 0.69 | 0.79 | 0.70 | 0.79 |
| ProGen2-large | 2.7B | CLM | 0.59 | 0.64 | 0.70 | 0.79 | 0.69 | 0.82 |
| ProtXLNet | 409M | PLM | 0.61 | 0.69 | 0.66 | 0.75 | 0.66 | 0.76 |
| ProtGPT2-C | 85M | CLM | 0.58 | 0.66 | 0.71 | 0.79 | 0.69 | 0.74 |
| ProtFIM (ours) | 85M | FIM | 0.57 | 0.71 | 0.74 | 0.82 | 0.74 | 0.81 |

Table 2: Model performances on SEIFER tasks in terms of precision.

| Model | #Params | Objective | H ($\alpha$-helix) | | E ($\beta$-sheet) | | C (Coil) | |
|---|---|---|---|---|---|---|---|---|
| | | | P@3 | P@5 | P@3 | P@5 | P@3 | P@5 |
| Random Generator | - | - | 0.25 | 0.25 | 0.49 | 0.47 | 0.47 | 0.45 |
| ProGen2-small | 151M | CLM | 0.32 | 0.32 | 0.45 | 0.43 | 0.47 | 0.48 |
| ProGen2-medium | 764M | CLM | 0.31 | 0.32 | 0.45 | 0.47 | 0.49 | 0.47 |
| ProGen2-large | 2.7B | CLM | 0.36 | 0.34 | 0.44 | 0.45 | 0.50 | 0.51 |
| ProtXLNet | 409M | PLM | 0.37 | 0.36 | 0.42 | 0.42 | 0.49 | 0.49 |
| ProtGPT2-C | 85M | CLM | 0.31 | 0.31 | 0.48 | 0.47 | 0.45 | 0.45 |
| ProtFIM (ours) | 80M | FIM | 0.31 | 0.32 | 0.45 | 0.46 | 0.48 | 0.48 |

models, such as ProGen2-small and ProGen2-large. These results prove that sequence design needs to be conducted using the surrounding context of target sites. On the other hand, our proposed ProtFIM shows comparable performance in term of retrieval and competitive performances in term of precision compared to other larger models by 2-20 times. These results show that the FIM scheme is parametrically efficient for protein middle engineering tasks.

## 4.4 PERFORMANCE WITH REGARD TO THE POSITION OF TARGET REGION

We start with an assumption that previous AR pLMs would be weak in FIM protein engineering because the sequence generation of AR pLMs is fulfilled by conditioning on only prefixes residues. To verify whether this phenomenon occurs, we ablate the SEIFER performance concerning the relative position of the target middle sites. After dividing each protein sequence into four parts, the $\alpha$-helix recovery performances of each model corresponding to each part are averaged and illustrated in Figure 3a and Figure 7a. Interestingly, in the first part (front part), only two models, ProtFIM and ProtXLNet, which consider both prefix and suffix part outperforms the random generator, while AR models such as ProtGPT2-C and ProGen-series do not. These results prove our assumption empirically.

Additionally, the fact that ProtFIM outperforms ProtXLNet in the front part shows the effectiveness of the FIM training scheme because ProtFIM has five times fewer parameters. Meanwhile, it is found that PLMs are generally better than the random generator in other parts, supporting the effectiveness of pLMs on protein middle engineering. In addition, it can be seen that the model's performance is not uniform over positions. We think that it is due to the lack of an evaluation dataset because the number of used CASP proteins is 28. However, since the models are compared under the same conditions, the insight obtained from the performance comparison in the experiments is reliable.

## 4.5 PERFORMANCE WITH REGARD TO LENGTH OF TARGET REGION

We can see that the random generator shows comparable performances to pLMs in several tests in the above results. To investigate this phenomenon, we ablate the SEIFER performances according to the length of the middle sites. We partition the range of lengths into four parts, and plot corresponding averaged Recall@K and Precision@K as in Figure 3b and Figure 7b. Interestingly, the random generator performs similarly to pLMs in the first quarter (short length size). However, the perfor-

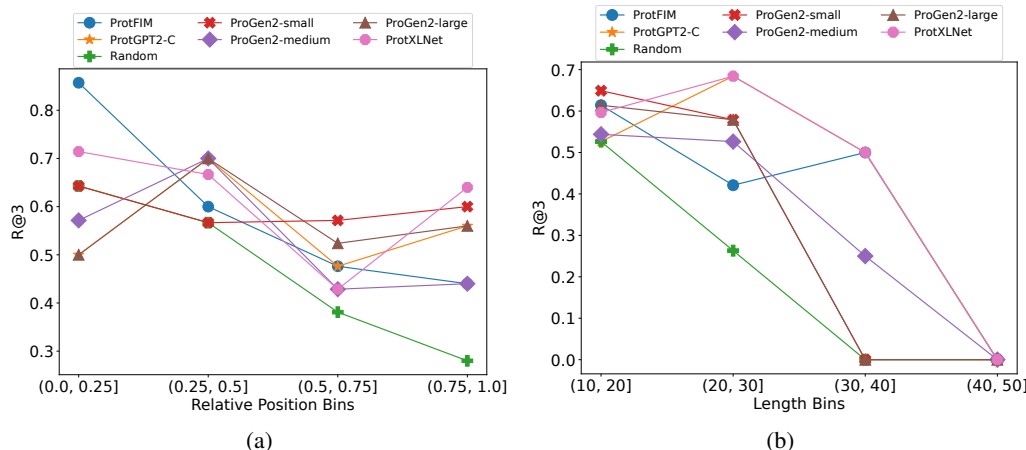

Figure 3: Performance changes in the term of retrieval with regard to (a) relative positions and (b) length of middle sites.

mance of the random generator drastically drops as the length of the target sites becomes longer, and it fails all predictions when the middle sites are longer than 30.

Meanwhile, the performance of pLMs degrades gradually and fails at the last part, where the middle sites are longer than 40. All the results imply that the length of the middle sites is the main factor for model performance. We explain this using the degree of freedom on possible protein structures of target middle sites. Since the high-dimensional interactions between amino acids make the structure of the protein, the structure is determined to some extent by the structural context from other residues except residues of the middle sites. In other words, the degree of freedom in the structure of the middle sites is relatively small due to the non-target residues. Considering that any amino acid is a building block of a $\alpha$-helix, $\beta$-sheet, and coil structure, even if the amino acid is randomly sampled, there will be a high probability of obtaining the desired original structure in FIM scenarios. Meanwhile, the observation that pLMs still work at the longer middle sites shows that pLMs would be a promising solution for long FIM protein sequence design, giving efficient sequence search compared to random generator.

## 5  ANALYSIS AND VISUALIZATION

Table 3: Zero-shot fitness prediction on FLIP tasks. All scores are Spearman correlation.

| Model | #Params | Objective | AAV | GB1 | Meltome | Meta Avg. |
|---|---|---|---|---|---|---|
| ESM-1b (mean) | 750M | MLM | 0.36 | 0.34 | 0.71 | 0.47 |
| ESM-1v (mean) | 750M | MLM | 0.33 | 0.38 | 0.72 | 0.48 |
| ProGen2-small | 151M | CLM | 0.39 | -0.21 | 0.56 | 0.25 |
| ProGen2-medium | 764M | CLM | 0.18 | -0.11 | 0.59 | 0.22 |
| ProGen2-large | 2.7B | CLM | 0.41 | 0.24 | 0.68 | 0.44 |
| ProtXLNet | 409M | PLM | 0.33 | 0.29 | 0.47 | 0.36 |
| ProtGPT2-C | 80M | CLM | 0.40 | 0.18 | 0.53 | 0.37 |
| ProtFIM | 80M | FIM | 0.39 | 0.25 | 0.60 | 0.41 |

### 5.1  REPRESENTATION QUALITY

Collecting experimental functional properties of protein sequence gives insights into a sequence-to-function relationship called fitness landscape. In protein engineering, the fitness landscape is used to rank designed sequences. To this end, pLMs can provide sequence representation for fitness prediction. Recently, FLIP benchmarks have been introduced to assess the quality of representations of pLMs (Dallago et al., 2021). Using FLIP, we compare the embeddings of ProtFIM with baselines.

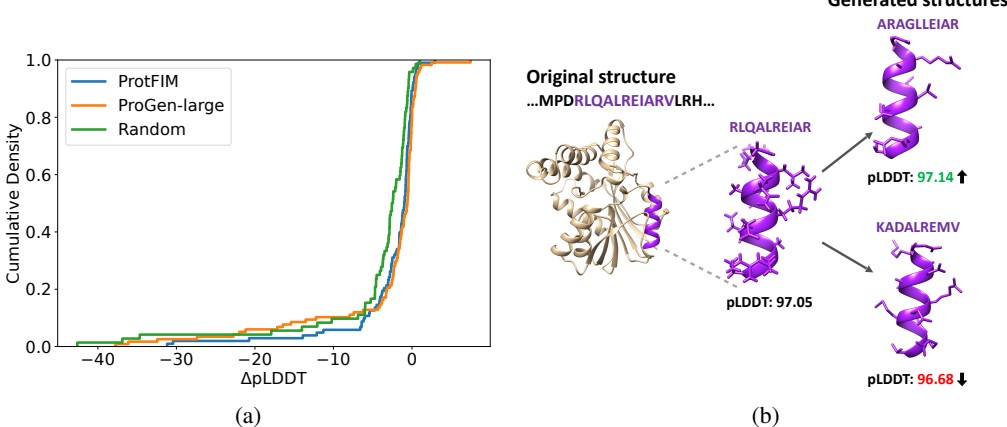

Figure 4: (a) Cumulative density plot on pLDDT change and (b) an example of a case where pLDDT increases or decreases after protein sequence design via ProtFIM.

Additionally, ESM-1b and 1v (Rao et al., 2020) are added to compare FIM with masked language modeling (MLM). In FLIP, embeddings of sequences are directly used to predict fitness without fine-tuning pLMs. For a fair comparison, embeddings are obtained via averaging of all residue representations. Table 3 shows that ProtFIM has comparable zero-shot fitness prediction performance even if the ProtFIM capacity is multiple times smaller than other models. This result implies that the embeddings of ProtFIM are effective for both FIM protein engineering and zero-shot fitness prediction. Detailed scores are included in Appendix A.8.

## 5.2 PLDDT CHANGE AND VISUALIZATION

AlphaFold2 gives a per-residue confidence metric called the predicted local distance difference test (pLDDT) ranging from 0 to 100. Recently, several works have used the metric as a scoring criterion to assess designed protein sequence by assuming that the higher pLDDT, the better and more plausible structure (Moffat et al., 2022; Wang et al., 2022). To assess the FIM engineering performance of models in terms of pLDDT, we visualize the difference between pLDDT of the structure of both new sequences and the corresponding original sequence using a cumulative density plot. Figure 4a reveals that positive cases where pLDDT increases after FIM engineering are rare for all models, but pLMs have more chance to get sequences with higher pLDDT. We cherry-pick a protein and visualize the original structure and modified structures through ProtFIM as shown in Figure 4b. The new two sequences of middle sites are different from the original sequences, but all have $\alpha$-helix. Interestingly, in-depth visualization considering the side-chain unveils the subtle difference, resulting in well or poorly-optimized sequences. All the above results demonstrate that our model, with the help of AlphaFold2, can serve as a sequence design framework, which optimizes the target sequence while maintaining the structures essential for the protein's function.

## 6 CONCLUSION

In this work, we show the FIM protein sequence design framework via pLMs and propose a new protein language model, ProtFIM, which is specialized for the framework. By evaluating various models via our proposed new evaluation scheme, SEIFER, ProtFIM performs FIM protein sequence design efficiently compared to existing pLMs. Additional analysis and visualization also prove that ProtFIM is a promising tool for practical protein engineering such as fitness prediction and sequence optimization.

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

# A    APPENDIX

## A.1    INTERACTION SITES EXTRACTION

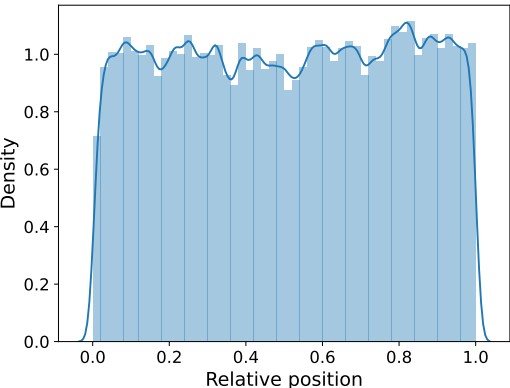

Figure 5: Relative positions of the interacting sites on the protein sequences.

We download 3D protein structures from PDB database (Sussman et al., 1998) and extract protein structures satisfying several conditions: having more than two protein chains; having UniProt ID and a length of the entire sequence in mmCIF dictionaries from MMCIF2Dict module of Biopython (Cock et al., 2009). We hypothesize that the two residue pairs of two different chains would be involved in the interaction if any atom excluding hydrogen of the residues were at a Euclidean distance of 8Å or less. Then, we identify all residues which are likely to be involved in the interactions and find where these residues are located on the entire protein sequence.

## A.2    SECONDARY STRUCTURE STATISTICS

We analyze the secondary structures of 166,512 structures that can be processed through a DSSP module of Biopython. Biopython classifies the secondary structures as eight classes by default: alpha helix (4-12) (code: 'H'), isolated beta-bridge residue (code: 'B'), strand (code: 'E'), 3-10 helix (code: 'G'), pi helix (code: 'I'), turn (code: 'T'), bend (code: 'S'), and none (code: '-'). In our study, The eight classes are mapped to the three classes as follows: 'H', 'G', and 'I' are mapped to the $\alpha$-helix class 'H'; 'B' and 'E' are mapped to the $\beta$-sheet class 'E'; 'T', 'S', 'C', and '-' are mapped to the coil class 'C'. In addition, in Figure 6b, '-' is displayed separately.

Figure 6a and 6b show that $\alpha$-helix substructures are dominant in natural proteins, meaning imbalance. Furthermore, coil structures have rules that are difficult to capture. Therefore, the model trained using the existing natural protein database would be familiar with the $\alpha$-helix generation. Therefore, a preprocessing or encoding technique that can alleviate the $\alpha$-helix bias can be a good research topic in the future.

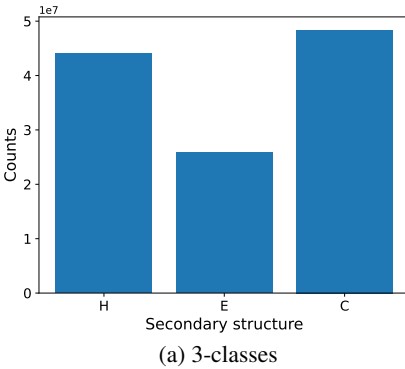
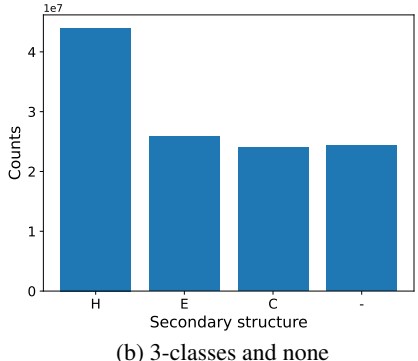

(a) 3-classes

(b) 3-classes and none

Figure 6: Secondary structure distribution of proteins in PDB database. H, E, C, and - correspond to $\alpha$-helix, $\beta$-sheet, coil, and none-type. (a) describes 3-classes secondary structure distribution. And, because coil can be divided into two categories, the coil and none-type structure in DSSP algorithm, we can calculate 4-classes distribution as shown in (b).

## A.3 TRAINING DATASETS

For training, protein sequences from UniRef50(Suzek et al., 2015) dated March 28, 2018 version are used to avoid leakage of CASP13, 14 and conduct a fair comparison with other models. 5% of protein sequences in the UniRef50 are randomly selected as a held-out validation set. The total number of sequences in training data is 25M.

## A.4 TRAINING DETAILS

ProtFIM is trained with a batch size of 128. The maximum length of each protein sequence we used for training is 1024. For ProtFIM optimization, we use AdamW optimizer Kingma & Ba (2014); Loshchilov & Hutter (2017) with a weight decay ratio of 1e-5. The learning rate is scheduled using cosine-warmup strategy. The total optimization step is 500k with 1k warmup steps. We train the model on 8 NVIDIA A100s in 4 days. FIM transformation is applied with 50% of probability. The model consists of 12 layers with a feature dimension of 768. The architecture is based on the released GPT2-base model by HuggingFace (Wolf et al., 2019).

## A.5 GENERATION HYPER-PARAMETERS

We conduct sequence generation using HuggingFace generation API. Th topK and topP values are set to 100 and 0.95. We set the temperature as 1.0. After sequence generation, we select top-K sequences. If shorter sequences are generated compared to the length of middle sites, we increase topK by 10 and conduct generation until K sentences are collected. We use the default option of HuggingFace API for other hyper-parameters. These hyper-parameters and generation processes are applied on ProtFIM, ProtGPT2, and ProGen2 models for a fair comparison. Also, we use ProtXLNet to generate sequences of target sites auto-regressively with conditioning on bidirectional context using topK sampling as other AR models.

## A.6 PERPLEXITY AND SEQUENCE RECOVERY

We also add other evaluation metrics, such as perplexity and sequence recovery rates, which are widely used for evaluating language models in inverse folding. Table 4 shows the result of perplexity and sequence recovery rates. ProtFIM performs poorly in terms of perplexity and sequence recovery rates. In the FIM paper (Bavarian et al., 2022b), some experiments find that perplexity alone is insufficient for evaluating the infilling task because infilling is conducted in a somewhat different nature compared to conventional left-to-light generation as expressed like $P_{FIM}(M \mid P, S) > P_{AR}(M \mid P)$ where P, M, S indicate prefix, middle, and suffix part, respectively.

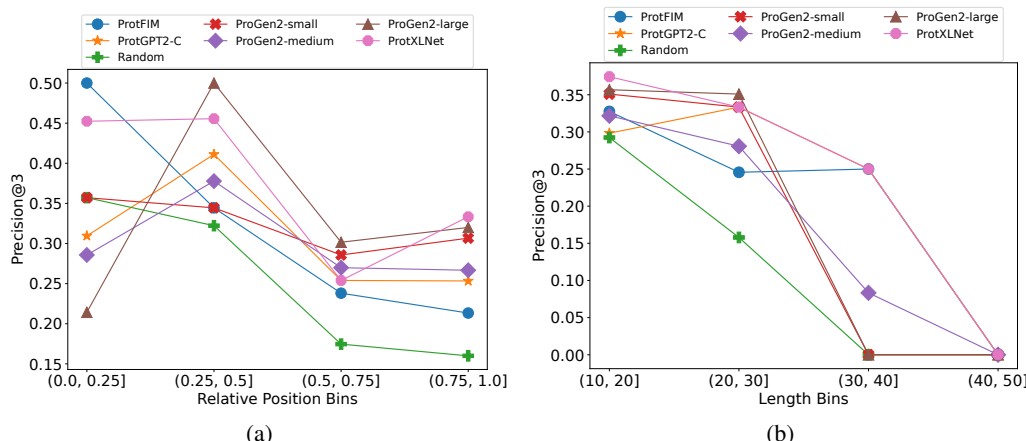

Figure 7: Performance changes in the metric of Precision@3 with regard to relative positions and length of middle sites.

Additionally, our targeted infilling task aims to design various sequences in the presence of local structure constraints considering the surrounding context, which is quite different from restoring residues as much as possible. So, there needs to be an appropriate evaluation scheme simulating protein middle engineering tasks that change amino acid residues of local parts of the protein to optimize the target protein, such as enzymes and antibodies. So, sequence residue recovery rate, a widely used metric to evaluate models' sequence design performance, is insufficient for the protein infilling task. Based on the above results and descriptions, we argue that our proposed SEIFER tasks are more appropriate for evaluating protein infilling tasks than existing metrics such as perplexity and sequence recovery rates.

Table 4: Perplexity and sequence recovery rates

| Model | #Params | Objective | Perplexity ($\downarrow$) | Recovery rate (%) ($\uparrow$) |
|---|---|---|---|---|
| ProGen2-small | 151M | CLM | 16.88 | 8 |
| ProGen2-medium | 764M | CLM | 16.17 | 10 |
| ProGen2-large | 2.7B | CLM | 16.24 | 9 |
| ProtXLNet | 409M | PLM | 16.58 | 8 |
| ProtGPT2-C | 80M | CLM | 17.08 | 8 |
| ProtFIM | 80M | FIM | 17.04 | 9 |

## A.7 PRECISION@K WITH REGARD TO POSITION AND LENGTH

Table 7a and 7b include ablation studies of SEIFER performance in term of precision according to relative positions and length of target sites in a protein.

## A.8 FLIP

Table 5, 6, and 7 contain the zero-shot fitness prediction performances of various pLMs on three fitness landscapes.

Table 5: Zero-shot fitness prediction on adeno-associated virus (AAV) capsid proteins (Bryant et al., 2021). All scores are Spearman correlation.

| Model | #Params | Objective | Mut-Des | Des-Mut | 1-vs-rest | 2-vs-rest | 7-vs-rest | low-vs-high | Avg. |
|---|---|---|---|---|---|---|---|---|---|
| ESM-1b (mean) | 750M | MLM | 0.63 | 0.59 | 0.04 | 0.26 | 0.46 | 0.18 | 0.36 |
| ESM-1v (mean) | 750M | MLM | 0.55 | 0.44 | 0.18 | 0.16 | 0.45 | 0.20 | 0.33 |
| ProGen2-small | 151M | CLM | 0.38 | 0.53 | 0.39 | 0.47 | 0.43 | 0.14 | 0.39 |
| ProGen2-medium | 764M | CLM | 0.19 | 0.25 | 0.14 | 0.30 | 0.22 | 0.00 | 0.18 |
| ProGen2-large | 2.7B | CLM | 0.68 | 0.67 | 0.33 | 0.20 | 0.42 | 0.13 | 0.41 |
| ProtXLNet | 409M | PLM | 0.55 | 0.58 | 0.21 | 0.02 | 0.42 | 0.20 | 0.33 |
| ProtGPT2-C | 80M | CLM | 0.59 | 0.66 | 0.24 | 0.34 | 0.41 | 0.16 | 0.40 |
| ProtFIM | 80M | FIM | 0.53 | 0.56 | 0.32 | 0.24 | 0.44 | 0.28 | 0.39 |

Table 6: Zero-shot fitness prediction on adeno-associated virus GB1 landscape (Wu et al., 2016). All scores are Spearman correlation.

| Model | #Params | Objective | 1-vs-rest | 2-vs-rest | 3-vs-rest | low-vs-high | Avg. |
|---|---|---|---|---|---|---|---|
| ESM-1b (mean) | 750M | MLM | 0.32 | 0.36 | 0.54 | 0.13 | 0.34 |
| ESM-1v (mean) | 750M | MLM | 0.32 | 0.32 | 0.77 | 0.10 | 0.38 |
| ProGen2-small | 151M | CLM | -0.27 | -0.30 | -0.26 | -0.03 | -0.21 |
| ProGen2-medium | 764M | CLM | -0.06 | -0.16 | -0.12 | -0.10 | -0.11 |
| ProGen2-large | 2.7B | CLM | 0.19 | 0.28 | 0.44 | 0.06 | 0.24 |
| ProtXLNet | 409M | PLM | 0.18 | 0.33 | 0.44 | 0.21 | 0.29 |
| ProtGPT2-C | 80M | CLM | 0.02 | 0.05 | 0.44 | 0.20 | 0.18 |
| ProtFIM | 80M | FIM | 0.01 | 0.18 | 0.63 | 0.18 | 0.25 |

Table 7: Zero-shot fitness prediction on landscape from the Meltome Atlas (Jarzab et al., 2020). All scores are Spearman correlation.

| Model | #Params | Objective | Mixed | Human | Human-Cell | Avg. |
|---|---|---|---|---|---|---|
| ESM-1b (mean) | 750M | MLM | 0.68 | 0.70 | 0.75 | 0.71 |
| ESM-1v (mean) | 750M | MLM | 0.67 | 0.75 | 0.74 | 0.72 |
| ProGen2-small | 151M | CLM | 0.46 | 0.63 | 0.59 | 0.56 |
| ProGen2-medium | 764M | CLM | 0.49 | 0.66 | 0.62 | 0.59 |
| ProGen2-large | 2.7B | CLM | 0.67 | 0.70 | 0.66 | 0.68 |
| ProtXLNet | 409M | PLM | 0.44 | 0.52 | 0.47 | 0.47 |
| ProtGPT2-C | 80M | CLM | 0.49 | 0.55 | 0.54 | 0.53 |
| ProtFIM | 80M | FIM | 0.51 | 0.66 | 0.63 | 0.60 |

