# OpenReview forum: "ProtFIM: Fill-in-Middle Protein Sequence Design via Protein Language Models"
_ICLR.cc/2023/Conference — Submitted to ICLR 2023_

### Official Review · Reviewer_ps6t · 2022-10-22

**Confidence:** 4
**Correctness:** 3
**Technical Novelty And Significance:** 2
**Empirical Novelty And Significance:** 1
**Recommendation:** 3

**Clarity, Quality, Novelty And Reproducibility:**

Clarity: The paper is generally clear, although the writing could be improved at places.

Quality: The empirical results are not the most convincing. Secondary structure conservation seems like a relatively easy task. What about sequence recovery or perplexity on the "middle" sequence? Fitness prediction results are also not showing improvements.

Originality: New training objective is the main new contribution, although I'm not sure if this is the best way to frame the training objective.



**Strength And Weaknesses:**

Strengths:
+ The "fill in the middle" use case is very common in protein engineering, and such a model could indeed be very useful.

Weaknesses:
+ The training setup choices seem somewhat limited: For example, it only allows one masked region in the middle. Often, in protein engineering, there might be multiple segments for joint design. There is a much more general version of this (e.g. see CM3 https://arxiv.org/pdf/2201.07520.pdf).
+ The empirical results on fitness prediction seem worse than existing methods.
+ I'm unconvinced that secondary structure conservation is the best evaluation scheme. What about sequence recovery or perplexity on the "middle" sequence?

**Summary Of The Paper:**

This paper introduced a new type of protein language model capable of filling in the middle of a sequence (which wasn't easily doable with autoregressive or with masked language models).

**Summary Of The Review:**

Important direction for supporting protein infilling with language models, but limited technical or empirical contributions.

---

> ### Author Response · Authors · 2022-11-18
> **Thank you. Our updates.**
>
> We thank the reviewer for the valuable feedback that helps us to improve our work.
> Please see our responses below:
>
> 1. **Training setup limitation on multiple segments target scenarios**
> - We really appreciate your pointing out the limitation and suggestions when conducting protein engineering on multiple segments. In this FIM scheme, the only possible solution to deal with the problem is iterative infilling. Namely, fill the first region and second region with filled residues on the first region and so on. As you suggested, CM3 would be the optimal solution to do this because CM3 utilizes a special mask token during masked causal modeling, and masked multiple segments can be filled during inference. Such multiple-segment infilling is also very important for enzyme or antibody design.
>
>
>     Meanwhile, we regard this work as a proof-of-concept work that first proposes an infilling formulation for protein engineering and proves the possibility of an infilling protein language model for the first time. Advanced infilling via causal masked language modeling like CM3 or other training setups can be considered as future work. Therefore, in the discussion section, we added sentences related to the future direction.
>
> 2. **Empirical results on fitness prediction seem worse than existing methods**
> - We added various protein language models that differ in pre-training objectives for the zero-shot prediction task. For a fair comparison, we use mean embedding of language models.
>
>     Compared with ProtGPT2-C [1], which is trained with the same number of parameters and data via CLM, ProtFIM has a better score. This proves that FIM learning provides both FIM ability and decent representation. In addition, in comparison with the ProGen series, ProtFIM has better performance than the 2x and 8x larger model (ProGen2-small, medium), or shows a slightly worse performance of 0.03 than the 30 times larger ProGen-large. Furthermore, ProtFIM has stronger representations than ProtXLNet [2], which can generate sequences with bidirectional context.
>
>     All generative models, including ProtFIM, have worse performance than the ESM series. However, the purpose of this experiment is not to achieve SOTA but to see how FIM modeling affects representation.
>
>     However, we argue that ProtFIM performs well, considering showing the third-best performance after ESM models despite eight times fewer parameters than ESM.
>
>     Based on all the results, we concluded that ProtFIM has good representation quality.
>
> 3. **Other metrics such as sequence recovery and perplexity**
> - Following the reviewer’s comments, we added perplexity and sequence recovery rates on each middle sequence as shown in appendix A.6.
>     On contrary to the SEIFER result, ProtFIM shows poor performance on both metrics. In the FIM paper, some experiments found that perplexity alone was insufficient for evaluating the infilling task because infilling is conducted in a somewhat different nature compared to conventional left-to-light generation as expressed like $P_{FIM}(middle|prefix, suffix) > P_{AR}(middle∣prefix)$.
>     Meanwhile, our targeted infilling task aims to design various sequences in the presence of local structure constraints considering the surrounding context, which is different from restoring residues as much as possible.
>     In fact, there needs to be a proper evaluation scheme simulating protein middle engineering tasks that change amino acid residues of local parts of the protein to optimize the target protein, such as enzymes and antibodies. Considering that objective is to design different protein sequences with optimized properties, sequence residue recovery rate, which is a widely used metric to evaluate the sequence design performance of models, is not sufficient for the protein infilling task.
>     Based on the above results and descriptions, we argue that our proposed SEIFER tasks are more appropriate for protein infilling tasks than existing metrics, such as perplexity and sequence recovery rate.
>
> [1] Noelia Ferruz, Steffen Schmidt, and Birte H ̈ocker. Protgpt2 is a deep unsupervised language model for protein design. Nature communications, 13(1):1–10, 2022.
>
> [2] Zhilin Yang, Zihang Dai, Yiming Yang, Jaime Carbonell, Russ R Salakhutdinov, and Quoc V Le. Xlnet: Generalized autoregressive pretraining for language understanding. Advances in neural information processing systems, 32, 2019b

---

### Official Review · Reviewer_c4rP · 2022-10-23

**Confidence:** 4
**Correctness:** 2
**Technical Novelty And Significance:** 2
**Empirical Novelty And Significance:** 2
**Recommendation:** 5

**Clarity, Quality, Novelty And Reproducibility:**

Clarity
The presentation is clear, but the choice of baselines prevents a clear understanding of the benefits of the presented approach.
Quality
The paper evaluates on appropriate datasets.
Novelty
Applying FIM for pLMs is novel.
Reproducibility
Evaluation was done on established benchmark sets and the work should be relatively easy to reproduce.


**Strength And Weaknesses:**

Strengths
- The FIM paradigm was not applied to pLMs so far.

Weaknesses
- No clear performance improvement over the ProGen baseline.
- The comparison to ProGen-large does not prove that FIM is more efficient per parameter. You should compare to an autogressive pLM of equal size without the FIM task to show that.
- If I understand correcty, also XLNet-style models should be capable of generating sequences with bidirectional context. Why is the ProtTrans XLNet model not considered?
- ESM-1b is not SOTA for zero-shot fitness prediction. Also, this evaluation again fails to prove that the parameter efficiency can be attributed to the FIM task.

Questions
- The definition of the @k metrics is hard to follow. Could it be expressed in a more straightforward way that directly explains what TP, FP, TN, FN are in the context?
- What is the reason for choosing a constructed secondary structure metric over perplexity? Perplexity can directly measure how well the model captures the masked span.
- Does the alpha helix bias mean that the FIM residue SS does not really depend much on the surrounding context?

Additional comments
In the introductory session, describing directed evolution as "random guessing or brute-force search" doesn't capture its essence, as it is actually meant to avoid doing just that.



**Summary Of The Paper:**

The paper proposes a protein language model (pLM) that is trained on a fill-in-middle task that has seen application in the NLP domain. This is motivated by the fact that existing pLMs that are used for protein generation are autoregressive, and as such cannot condition on bidirectional context, which would actually be the most prominent case when re-engineering a region of a protein, as many residues relevant to function are found in the middle of the sequence rather than at its ends. The performance of the infilling model is evaluated by predicting the secondary structure of the filled in region and comparing it to the ground truth.

**Summary Of The Review:**

The paper introduces a FIM-trained pLM and argues in favor of it's performance based on the fact that it has a far lower number of parameters than the ProGen model used for comparison. However, this leaves it unclear whether the model is truly more parameter efficient because of the FIM task - an AR pLM of equal size would be required to show that. Also, additional baseline methods should be considered to convincingly show that FIM boosts performance for the investigated tasks.

---

> ### Author Response · Authors · 2022-11-18
> **Thank you. Our updates.**
>
> We thank the reviewer for the valuable feedback that helps us to improve our work.
> Please see our responses below:
>
> 1. **No clear performance improvement**
> - To clarify the parametric efficiency of ProtFIM, more models of ProGen2 with different parameters (150M, 764M, 2.6B) were added. Interestingly, 85M-sized ProtFIM has similar or comparable performances in SEIFER compared to ProGen2 models.
> Specially, we note that only ProtFIM visibly outperforms all other models in the SEIFER task when optimizing the front part of the target protein, as shown in figure 3(a). In contrast, ProGen2 models with billions of parameters have similar performance at the level of the random generator.
>     All the newly added results demonstrate ProtFIM's parameter efficiency in protein infilling task.
>
> 2. **Comparison to auto-regressive pLMs of equal size without the FIM task**
> - Following the reviewer’s comments, we added more comparisons with other LM works to set a concrete experimental basis. First, we newly trained and added a GPT2-style protein language model, ProtGPT2-C [1]. ProtGPT2-C has the same number of parameters and learns the same data as ProtFIM. Therefore, we believe the ProtGPT2-C gives a fair comparison to prove the FIM scheme compared to conventional causal language modeling (CLM) in the infilling task. By evaluating Both SEIFER and other embedding tasks, we demonstrate that the ProtFIM performs better than ProtGPT2-C. The result means that Fill-in-Middle ability after FIM learning is effective for protein middle sequence engineering.
>
>     A more detailed analysis in figure 3(a) showed that ProtFIM is about 25% better than ProtGPT2 when fixing the front end of the target protein. In particular, ProtGPT2 produces worse results than random generators.
>
>     These results prove that ProtFIM has fill-in-middle ability in addition to the generation performance of ProtGPT2-C.
>
> 3. **Comparison to other LM models (XLNet-style models)**
> - We thank the reviewer for pointing out the XLNet-style models that we missed. We agree that XLNet is a promising option to conduct fill-in-middle engineering using both the prefix and suffix. We added the officially released ProtXLNet and conducted the same experiments on the model. Comprehensive results show that ProtXLNet has a strong performance in SEIFER tasks compared to the similar scale of models, such as ProGen2-small and ProGen2-medium. These results are another finding supporting that Fill-in-middle sequence design needs to be conducted using the surrounding context of target sites.
>     Additionally, we note that only ProtFIM visibly outperforms the ProtXLNet in SEIFER task when optimizing the front part of the target protein as shown in figure 3(a).
>
> 4. **Protein fitness prediction**
> - We followed the reviewer's comment and added various protein language models that differ in pre-training objectives for the zero-shot prediction task. For a fair comparison, we use mean embedding of language models.
>
>     Compared with ProtGPT2-C, which is trained with the same number of parameters and data via CLM, ProtFIM has a better score. This proves that FIM learning provides both FIM ability and decent representation. In addition, in comparison with the ProGen series, ProtFIM has better performance than the 2x, 8x, and 80x larger model (ProGen2-small, medium, and xlarge), or shows a slightly worse performance of 0.03 than the 30 times larger ProGen-large. Furthermore, ProtFIM has stronger representations than ProtXLNet, which can generate sequences with bidirectional context.
>
>     All generative models, including ProtFIM, have worse performance than the ESM series. However, the purpose of this experiment is not to achieve SOTA but to see how FIM modeling affects representation. However, we argue that ProtFIM performs well, considering showing the third-best performance after ESM models despite eight times fewer parameters than ESM. Based on all the results, we concluded that ProtFIM has good representation quality.

---

> > ### Author Response · Authors · 2022-11-18
> > **Thank you. Our updates.**
> >
> > 5. **The definition of the @k metric**
> > - In the SEIFER task, all models generate K sequences for N middle sites, and all sequences are evaluated by whether the whole secondary structure at each site is recovered. If the whole secondary structure is recovered, it is a true positive. Then, Precision@K is mean(TP/K). We wanted to evaluate retrieval performance, but we recognized that Recall@K would be confusing after considering the reviewer’s comment. Retrieval@K equals one if the true positive sequence exists in generated K sequences. Thus, Retrieval@K is (the number of sites having TP among K)/N.
> >
> > 6. **Choosing a secondary structure metric instead of perplexity**
> > - Following the reviewer’s comments, we added perplexity to each middle sequence in Appendix A.6..  On the contrary to the SEIFER result, ProtFIM shows poor performance on both metrics. In the FIM paper
> >
> >     In the FIM paper, some experiments found that perplexity alone was insufficient for evaluating the infilling task because infilling is conducted in a somewhat different nature compared to conventional left-to-light generation as expressed like $P_{FIM}(middle|prefix, suffix) > P_{AR}(middle∣prefix)$.
> >
> >     Meanwhile, our targeted infilling task aims to design various sequences in the presence of local structure constraints considering the surrounding context, which is different from restoring residues as much as possible.
> >
> >     In fact, there needs to be a proper evaluation scheme simulating protein middle engineering tasks that change amino acid residues of local parts of the protein to optimize the target protein, such as enzymes and antibodies. Considering that objective is to design different protein sequences with optimized properties, sequence residue recovery rate, which is a widely used metric to evaluate the sequence design performance of models, is not sufficient for the protein infilling task.
> >
> >     Based on the above results and descriptions, we argue that our proposed SEIFER tasks are more appropriate for protein infilling tasks than existing metrics, such as perplexity.
> >
> > 7. **Alpha helix bias**
> > - The alpha helix bias we are talking about does not simply mean that the model outputs the residues of the alpha helix without considering the context. The FIM model generates a sequence based on the context before and after the target region as expressed in P_FIM(middle|prefix, suffix).
> >
> >     The alpha helix bias we mentioned means that models have more possibility to be trained to be specialized at generating residues of the alpha helix.
> >
> >     As shown in appendix figure 1, alpha helix substructures are dominant in natural proteins, meaning imbalance compared to other local structures, such as coils and sheets. Furthermore, coil structures, which have randomized folded structures, have rules that are difficult to capture.
> >
> >     Therefore, the model trained from the existing natural protein database is likely to be trained to be familiar with the alpha helix generation.
> >     A unique preprocessing or encoding technique that can alleviate the alpha helix bias can be a good research topic in the future.
> >
> > 8. **Random guessing and brute-force search**
> > - We thank the reviewer's comment and we remove these expressions.
> >
> > [1] Noelia Ferruz, Steffen Schmidt, and Birte H ̈ocker. Protgpt2 is a deep unsupervised language model for protein design. Nature communications, 13(1):1–10, 2022.

---

### Official Review · Reviewer_tRtc · 2022-10-25

**Confidence:** 4
**Correctness:** 3
**Technical Novelty And Significance:** 2
**Empirical Novelty And Significance:** 2
**Recommendation:** 5

**Clarity, Quality, Novelty And Reproducibility:**

 The article is easy to understand. The novelty is marginal.  ProtFIM achieves similar performance to Progen-large, but ProFIM has less parameters.

**Details Of Ethics Concerns:**

No ethics concerns.

**Strength And Weaknesses:**

Strengths:
1.	The article is well written and easy to understand.

Weaknesses:

 1. There are too few metrics for experimental evaluation. Only P@k and R@k are included in evaluating infilling.  Some metrics provided in Progen2 can be used (e.g. complexity).

 2. The method is not throughly evaluated.  The number of comparison methods is too small. There are still a lot of related work on generative PLMs.   The performance of the model is similar to Progen-large, and the main advantage is that the model size is smaller. Is it possible to add other models with different scales provided in Progen, especially small-scale models?  Then we can see whether ProtFIM still has an advantage over the models sharing similar scales.

3. The evaluation requires the consistence of secondary structure between the generated protein and the original protein. However, a new protein with better performance may have a different secondary structure. Whether asking the generated protein is completely similar to the original one in secondary structure will affect the innovation of the model?

4. It’s better to explain what (a) and (b) are in the caption of Figures 3 and 4. In addition, Figure 4 (a), (b) are both “Precision”.


**Summary Of The Paper:**

The authors proposed a new protein language model ProtFIM, and a new protein sequence design framework via ProtFIM.
By comparing the performance with previous models via their new evaluation scheme ( SEIFER), ProtFIM achieved similar performance but with less parameters.

**Summary Of The Review:**

The idea of this paper is easy to understand, but not novel enough.  The performance of ProtFIM is similar to Progen-large, but ProtFIM has less parameters. More competing methods and some other metrics should be included.

---

> ### Author Response · Authors · 2022-11-18
> **Thank you. Our updates.**
>
> We thank the reviewer for the valuable feedback that helps us to improve our work.
> Please see our responses below:
>
> 1. **Lack of metrics for evaluation**
>
> - Following the reviewer’s comments, we added perplexity and sequence recovery rates widely used for evaluating protein language models and inverse folding, respectively as shown in Appendix A.6. On the contrary to the SEIFER result, ProtFIM shows poor performance on both metrics. In the FIM paper, some experiments found that perplexity alone was insufficient for evaluating the infilling task. In addition, our targeted infilling task aims to design various sequences in the presence of local structure constraints considering the surrounding context, which is different from restoring residues as much as possible.
> Based on the two facts, we argue that SEIFER is more appropriate than the existing metrics for protein infilling tasks.
>
> 2. **Comparison to other LM works**
> - Following the reviewer’s comments, we added more comparisons with other LM works to set a concrete experimental basis. First, we newly trained and added a GPT2-style protein language model, ProtGPT2-C [1]. ProtGPT2-C has the same number of parameters and learns the same data as ProtFIM. Therefore, we believe the ProtGPT2-C gives a fair comparison to prove the FIM scheme compared to conventional causal language modeling (CLM) in the infilling task. By evaluating Both SEIFER and other embedding tasks, we demonstrate that ProtFIM performs better than ProtGPT2-C. The result means that the Fill-in-Middle ability after FIM learning is effective for protein middle sequence engineering.
>
>     Second, we conducted the same experiments on the officially released ProtXLNet ([https://github.com/agemagician/ProtTrans](https://github.com/agemagician/ProtTrans)). ProtXLNet [2] was optimized via permutation language modeling in which all permutations are fed into the model during auto-regressive modeling. So, this model is another option that is able to fill-in-middle protein engineering using both the prefix and suffix. Comprehensive results show that ProtXLNet has a strong performance in SEIFER tasks compared to the similar scale of models, such as ProGen2-small and ProGen2-medium. These results are another finding supporting that Fill-in-middle sequence design needs to be conducted using the surrounding context of target sites.
>
>     Meanwhile, as a representative of scaled CLM, more models of ProGen2 with different parameters (150M, 764M, and 2.6B) were added. Interestingly, 85M-sized ProtFIM has comparable performances in SEIFER compared to ProGen2 models. Especially, we note that only ProtFIM visibly outperforms all other models in the SEIFER task when optimizing the front part of the target protein, as shown in figure 3(a). In contrast, ProGen2 models with billions of parameters have similar performance at the level of the random generator.
>
> 3. **Validity of our evaluation using secondary structure**
> - Protein engineering, which optimizes the protein’s function by changing amino acid residues of local parts of protein, has been a widely used technique for designing enzymes, antibodies, pharmaceuticals, and agriculture.
> A representative technique of protein engineering is mutagenesis, which conducts mutations on amino acid residues via genetic engineering. However, mutation search space explodes if the target sites are longer, resulting in substantial efforts in high-throughput screening experiments.
> Protein Generative language models would be a promising alternative for an artificial mutation engineer relying on evolutionary knowledge from causal language modeling. Sequence residue recovery has been widely used to evaluate the generation performance of Protein Generative language models.  However, we argue that recovery of whole residues are not proper because our objective of sequence optimization is to design different protein sequence with optimized properties.
> So, a metric is needed to evaluate whether the model can generate a variety of sequences while retaining the function of the protein. Because the function of protein is directly linked to local structure, evaluating the model capacity that generates different residues with the same local structure is a promising way. Actually, optimizing local structure with different residues are widely used way to optimize protein sequence for protein engineering.
> So, we argue that our proposed SEIFER tasks are appropriate for simulating protein sequence engineering, because in SEIFER tasks models are tasked to recover the protein's local structure, especially, secondary structures, not residues.
> We added “Difference of SEIFER over protein residue recovery task” paragraph in Section 4.1 to include the above explanation.
>
>
> 4. **Modifying figure 3 and 4**
> - We modify the captions for (a) and (b) of Figure 3 to be more clarified. Also, we correct the typo in Figure 4.

---

> > ### Author Response · Authors · 2022-11-18
> > **Thank you. Our updates.**
> >
> > [1] Noelia Ferruz, Steffen Schmidt, and Birte H ̈ocker. Protgpt2 is a deep unsupervised language model for protein design. Nature communications, 13(1):1–10, 2022.
> >
> > [2] Zhilin Yang, Zihang Dai, Yiming Yang, Jaime Carbonell, Russ R Salakhutdinov, and Quoc V Le. Xlnet: Generalized autoregressive pretraining for language understanding. Advances in neural information processing systems, 32, 2019b.

---

### Official Review · Reviewer_2Lrn · 2022-10-31

**Confidence:** 4
**Correctness:** 4
**Technical Novelty And Significance:** 1
**Empirical Novelty And Significance:** 2
**Recommendation:** 5

**Clarity, Quality, Novelty And Reproducibility:**

The paper can be improved in clarity in general. While the idea of the paper is straightforward and thus easy to understand. The organization of the content, the notation, and figures can be revised to improve the flow. For instance, the introduction section can become sharper and more concise while the method section can benefit from more formal and structured content of the general problem, possibly the existing variants and the proposed approach.



**Strength And Weaknesses:**

- the paper provides a novel method for PLM especially suitable for sequence-based protein optimization.
- despite being a language model, the proposed architecture is relatively small in number of parameters which enables efficient use at test time and enables research for protein-engineering with limited computational resources.
- the additional experiments regarding the quality and the general analysis of the learnt representations are interesting, encouraging and informative.

*Weaknesses*:
- For an ML conference there seems to be no technical novelty, neither fundamental, nor incremental. The model is almost an exact copy of the standard LM with the addition of the structure constraint.
- The number of available protein (and non protein) language models are vast. The paper only compares with one method while it could (and should have) compared with other PLMs but also possibly other language models with the same trick (moving the missing part to the end).
- the main results that is the goal of the work for protein engineering seem quite comparable with the only baseline that is used (ProGen).

**Summary Of The Paper:**

The paper's aim is sequence-based protein engineering based on protein language models (PLMs). For this purpose, it uses a recently developed self-supervised in-filling language model. The model rearranges the middle (to be infilled) part of a sequence and move it to the end of the sequence which enables the use of standard forward prediction. The paper then tests these PLM models for identifying other possible residue sequences in the middle of a protein sequence that preserves the general 3D structure of the protein (secondary structural type).

**Summary Of The Review:**

The paper uses a novel method for protein language models that is quite efficient in number of parameters and shows the results are comparable for protein sequence engineering compared to one other recent PLM baseline. However, considering that the results are clearly positive compared to the possible baselines, the lack of technical novelty (even in form of an increment on existing methods), the lack of a thorough set of baselines, and presentation of the work, I believe the paper is not ready for publication, at least at a ML venue. More developments on the method and/or a more thorough empirical evidence will increase the quality of the paper in another revision.

---

> ### Author Response · Authors · 2022-11-18
> **Thank you. Our updates.**
>
> We thank the reviewer for the valuable feedback that helps us to improve our work.
> Please see our responses below:
>
> 1. **The novelty of our work**
> - In our humble view, first, our contribution is to revisit the protein local sequence optimization task, which is a key element in protein engineering, as a protein infilling task using protein language modeling. And through results, we showed that efficient protein local sequence optimization can be realized via the appropriate machine technique, Fill-in-middle sequence generation, to supplement the existing domain conventional method. We believe this contribution is enough to be assessed in Machine Learning for Sciences (eg biology, physics, health sciences, social sciences, climate/sustainability ) venue.
>
>     Second, we proposed a novel evaluation scheme designed for protein local sequence engineering. There needs to be a proper evaluation scheme simulating protein middle engineering tasks that change amino acid residues of local parts of the protein to optimize the target protein, such as enzymes and antibodies. Considering that objective is to design different protein sequences with optimized properties, sequence residue recovery rate, which is a widely used metric to evaluate the sequence design performance of models, is not sufficient for the protein infilling task.
>
> 2. **Comparison to other LM works**
> - Following the reviewer’s comments, we added more comparisons with other LM works to set a concrete experimental basis. First, we newly trained and added a GPT2-style protein language model, ProtGPT2-C [1]. ProtGPT2-C has the same number of parameters and learns the same data as ProtFIM. Therefore, we believe the ProtGPT2-C gives a fair comparison to prove the FIM scheme compared to conventional causal language modeling (CLM) in the infilling task. By evaluating Both SEIFER and other embedding tasks, we demonstrate that ProtFIM performs better than ProtGPT2-C. The result means that the Fill-in-Middle ability after FIM learning is effective for protein middle sequence engineering.
>
>     Second, we conducted the same experiments on the officially released ProtXLNet ([https://github.com/agemagician/ProtTrans](https://github.com/agemagician/ProtTrans)). ProtXLNet [2] was optimized via permutation language modeling in which all permutations are fed into the model during auto-regressive modeling. So, this model is another option that is able to fill-in-middle protein engineering using both the prefix and suffix. Comprehensive results show that ProtXLNet has a strong performance in SEIFER tasks compared to the similar scale of models, such as ProGen2-small and ProGen2-medium. These results are another finding supporting that Fill-in-middle sequence design needs to be conducted using the surrounding context of target sites.
>
>     Meanwhile, as a representative of scaled CLM, more models of ProGen2 with different parameters (150M, 764M, and 2.6B) were added. Interestingly, 85M-sized ProtFIM has comparable performances in SEIFER compared to ProGen2 models. Especially, we note that only ProtFIM visibly outperforms all other models in the SEIFER task when optimizing the front part of the target protein, as shown in figure 3(a). In contrast, ProGen2 models with billions of parameters have similar performance at the level of the random generator.
>
>
> 3. **Clarity**
> - We modify our paper to increase clarity (e.g. the introduction more concisely).
>
> [1] Noelia Ferruz, Steffen Schmidt, and Birte H ̈ocker. Protgpt2 is a deep unsupervised language model for protein design. Nature communications, 13(1):1–10, 2022.
>
> [2] Zhilin Yang, Zihang Dai, Yiming Yang, Jaime Carbonell, Russ R Salakhutdinov, and Quoc V Le. Xlnet: Generalized autoregressive pretraining for language understanding. Advances in neural information processing systems, 32, 2019b.

---

### Author Response · Authors · 2022-11-18
**General and experimental updates and comments**

# Summary of revisions

We thank all reviewers for taking the time to provide detailed comments and to suggest additional studies for our paper. Our primary responses to each issue raised by the reviewer are written in the response to each review. Here, we summarize the major changes in our submission.

# General updates and comments

- **Validity of our evaluation using secondary structure**
    - Protein engineering, which optimizes the protein’s function by changing amino acid residues of local parts of protein, has been a widely used technique for designing enzymes, antibodies, pharmaceuticals, and agriculture.
    A representative technique of protein engineering is mutagenesis, which conducts mutations on amino acid residues via genetic engineering. However, mutation search space explodes if the target sites are longer, resulting in substantial efforts in high-throughput screening experiments.
    Protein Generative language models would be a promising alternative for an artificial mutation engineer relying on evolutionary knowledge from causal language modeling. Sequence residue recovery has been widely used to evaluate the generation performance of Protein Generative language models.  However, we argue that recovery of whole residues are not proper because our objective of sequence optimization is to design different protein sequence with optimized properties.
    So, a metric is needed to evaluate whether the model can generate a variety of sequences while retaining the function of the protein. Because the function of protein is directly linked to local structure, evaluating the model capacity that generates different residues with the same local structure is a promising way. Actually, optimizing local structure with different residues are widely used way to optimize protein sequence for protein engineering.
    So, we argue that our proposed SEIFER tasks are appropriate for simulating protein sequence engineering, because in SEIFER tasks models are tasked to recover the protein's local structure, especially, secondary structures, not residues.
    We added “Difference of SEIFER over protein residue recovery task” paragraph in Section 4.1 to include the above explanation.
- **Clarifying our paper**
    - As the reviewers suggest, we modify the introduction concisely, clarify k@metric, and correct typos.

# ****Experimental improvements and updates****

- **Adding other LM baselines (ProtGPT2-C, ProtXLNet, and ProGen2 with different parameters)**
    - Following the reviewer’s comments, we added more comparisons with other LM works to set a concrete experimental basis. First, we newly trained and added a GPT2-style protein language model, ProtGPT2-C [1]. ProtGPT2-C has the same number of parameters and learns the same data as ProtFIM. Therefore, we believe the ProtGPT2-C gives a fair comparison to prove the FIM scheme compared to conventional causal language modeling (CLM) in the infilling task. By evaluating Both SEIFER and other embedding tasks, we demonstrate that ProtFIM performs better than ProtGPT2-C. The result means that the Fill-in-Middle ability after FIM learning is effective for protein middle sequence engineering.
    Second, we conducted the same experiments on the officially released ProtXLNet ([https://github.com/agemagician/ProtTrans](https://github.com/agemagician/ProtTrans)). ProtXLNet [2] was optimized via permutation language modeling in which all permutations are fed into the model during auto-regressive modeling. So, this model is another option that is able to fill-in-middle protein engineering using both the prefix and suffix. Comprehensive results show that ProtXLNet has a strong performance in SEIFER tasks compared to the similar scale of models, such as ProGen2-small and ProGen2-medium. These results are another finding supporting that Fill-in-middle sequence design needs to be conducted using the surrounding context of target sites.
    Meanwhile, as a representative of scaled CLM, more models of ProGen2 with different parameters (150M, 764M, and 2.6B) were added. Interestingly, 85M-sized ProtFIM has comparable performances in SEIFER compared to ProGen2 models. Especially, we note that only ProtFIM visibly outperforms all other models in the SEIFER task when optimizing the front part of the target protein, as shown in figure 3(a). In contrast, ProGen2 models with billions of parameters have similar performance at the level of the random generator.

---

> ### Author Response · Authors · 2022-11-18
> **General and experimental updates and comments**
>
> - **Adding other metrics for evaluation (sequence recovery and perplexity)**
>     - Following the reviewer’s comments, we added perplexity and sequence recovery rates widely used for evaluating protein language models and inverse folding, respectively as shown in Appendix A.6. On the contrary to the SEIFER result, ProtFIM shows poor performance on both metrics. In the FIM paper, some experiments found that perplexity alone was insufficient for evaluating the infilling task. In addition, our targeted infilling task aims to design various sequences in the presence of local structure constraints considering the surrounding context, which is different from restoring residues as much as possible.
>     Based on the two facts, we argue that SEIFER is more appropriate than the existing metrics for protein infilling tasks.
> - **Improvement in representation quality analysis (zero-shot fitness prediction)**
>     - We followed the reviewer's comment and added various protein language models that differ in pre-training objectives for the zero-shot prediction task. For a fair comparison, we use mean embedding of language models. Compared with ProtGPT2-C, which is trained with the same number of parameters and data via CLM, ProtFIM has a better score. This proves that FIM learning provides both FIM ability and decent representation. In addition, in comparison with the ProGen series, ProtFIM has better performance than the 2x and 8x larger models (ProGen2-small and medium) or shows a slightly worse performance of 0.03 than the 30 times larger ProGen-large. Furthermore, ProtFIM has stronger representations than ProtXLNet, which can generate sequences with bidirectional context.
> All generative models, including ProtFIM, have worse performance than the ESM series. The purpose of this experiment is not to achieve SOTA but to see how FIM modeling affects representation. However, we argue that ProtFIM performs well, considering showing the third-best performance after ESM models despite eight times fewer parameters than ESM. Based on all the results, we concluded that ProtFIM has good representation quality. Additionally, to be convenient for fitness prediction comparison, we summarized three fitness landscapes by averaging all scores and added a new table (table 3) containing the result.  Previously detailed scores are moved to appendix A.8.
> - **Updating the minimum length of beta-sheet and coil**
>     - In the previous version, we use 10, 4, and 4 for a minimum length of residues of the helix, sheet, and coil secondary structure for SEIFER task evaluation. In the new updated version, we use 10, 6, and 6 to simulate the more practical environment of secondary structures [4].
>
> [1] Noelia Ferruz, Steffen Schmidt, and Birte H ̈ocker. Protgpt2 is a deep unsupervised language model for protein design. Nature communications, 13(1):1–10, 2022.
>
> [2] Zhilin Yang, Zihang Dai, Yiming Yang, Jaime Carbonell, Russ R Salakhutdinov, and Quoc V Le.
>
> Xlnet: Generalized autoregressive pretraining for language understanding. Advances in neural
>
> information processing systems, 32, 2019b.
>
> [3] Mohammad Bavarian, Heewoo Jun, Nikolas Tezak, John Schulman, Christine McLeavey, Jerry
> Tworek, and Mark Chen. Efficient training of language models to fill in the middle. arXiv
> preprint arXiv:2207.14255, 2022a.
>
> [4] [https://www.cryst.bbk.ac.uk/PPS2/course/section8/ss-960531_10.html](https://www.cryst.bbk.ac.uk/PPS2/course/section8/ss-960531_10.html)

---

### Author Response · Authors · 2022-12-08
**Revised paper uploaded and response to reviewers' comments posted**

Dear Reviewers:

We would like to thank you for your constructive comments and suggestions, which are very helpful for improving our paper. We have posted point-to-point reply to each question/comment raised by you and uploaded the revised version of our paper (with track changes marked in red). Please do feel free to let us know if you have any further questions.

Thank you very much.

Best regards,

The Authors of the Paper

---

### Author Response · Authors · 2022-12-12
**Kind reminder**

Dear Reviewers:


We would like to thank you for your constructive comments and suggestions, which are very helpful for improving our paper.
As we noticed, we have posted a point-to-point reply to each question/comment raised by you and uploaded the revised version of our paper (with track changes marked in red).
We would really appreciate it if all the reviewers and ACs consider our revisions during discussion periods.
Please do feel free to let us know if you have any further questions.

Thank you very much.


Best regards,

The Authors of the Paper

---

### Decision · Program_Chairs · 2023-01-20

**Decision:**

Reject

**Justification For Why Not Higher Score:**

The number of available protein (and nonprotein) language models is vast. The paper only compares the new method to one existing approach while it could (and should have) compared with other protein language models and possibly other language models with the same trick (moving the missing part to the end).

**Justification For Why Not Lower Score:**

N/A

**Metareview: Summary, Strengths And Weaknesses:**

The paper describes a protein language model to support sequence-based protein engineering. It uses a self-supervised in-filling language model that rearranges the middle (to be infilled) part of a sequence and moves it to the end of the sequence, enabling standard forward prediction. The paper then tests these models to identify other possible residue sequences in the middle of a protein sequence that preserves the general 3D structure of the protein (secondary structural type).

**Strengths:**

* The article is well-written and easy to understand.
* Experiments regarding the quality and the general analysis of the learned representations are interesting and informative.

**Weaknesses:**

* Reviewers agree there seems to be very little technical novelty and that the current approach is neither fundamental nor incremental. The model is almost an exact copy of the standard LM with the addition of the structure constraint.
* The method is not thoroughly evaluated. The number of comparison methods is too small. Few performance metrics are considered in experimental evaluation.
* There is also consensus that the approach does not provide any clear performance improvement over the ProGen baseline. The empirical results on fitness prediction seem worse than existing methods.